# Mechanical properties and energy damage evolution mechanism of fiber-reinforced cemented sulfur tailings backfill under uniaxial compression

**Wei Liu[1], Yongqiang Hou[1], Shenghua Yin[1]\*, Yanli Wang[2], Huihui Du[1], Minzhe Zhang[1]**

**1** School of Civil and Resource Engineering, University of Science and Technology Beijing, Beijing, China,
**2** The 52nd Research Institute of China Ordnance Industry Group, Yantai, China

\* hyq2428204462@163.com

**Data Availability Statement:** All relevant data are within the paper and its Supporting Information files.

## Abstract

This paper studies mechanical properties and energy damage evolution of fiber-reinforced cemented sulfur tailings (CSTB) backfill. The effects of fiber length and fiber content on the stress, toughness and failure properties of the CSTB were systematically revealed. In addition, the energy index evolution law was studied, and the energy damage evolution mechanism of CSTB was revealed. The results show that the deformation failure of fiber-reinforced CSTB mainly goes through four stages: initial crack compaction, linear elastic deformation, yield failure and post-peak failure. The peak stress and residual stress of the CSTB firstly increase and then decrease with the increase of fiber content and the addition of fiber can promote the change from brittle failure to ductile failure of the CSTB. Adding appropriate amount of fiber can improve the toughness of CSTB, and the influence degree of fiber length on the toughness index of CSTB is 6mm>12mm>3mm. The total strain energy increases linearly along the variation of fiber content, while the elastic strain energy and dissipated energy increase exponentially at the peak stress point. In the process of CSTB deformation and failure, "gentle—linear growth—slow growth—rapid decline" is for elastic strain energy, while "gentle—slow growth—rapid growth—linear growth" is for dissipation energy. The damage and failure of CSTB mainly experienced four stages: initial damage, slow growth of damage, accelerated damage and damage failure, and the damage evolution curve also showed the changing characteristics of "gentle—slow growth—rapid growth—linear growth". The CSTB without added fiber showed obvious "Y-type" and "linear-type" shear failure characteristics and the phenomenon of shear cracks penetrating the backfill appeared. No big shear crack occur when it is damaged, showing that the fiber addition restrain the crack growth and improve the overall crack resistance of the CSTB. Hydration products are obviously distributed on the surface of the fiber, which indicates that the fiber will be evenly dispersed in the CSTB and form a certain bonding force with the cement-tailings matrix, thus improving the overall mechanical properties of the CSTB.

**Funding:** This work was supported the Shandong Provincial Major Science and Technology Innovation Project, China (No. 2019SDZY05), the Key Program of National Natural Science Foundation of China (Grant No. 51734001), the Key Program of National Natural Science Foundation of China (52034001). The funders had no role in study design, data collection and analysis, decision to publish, or preparation of the manuscript.

**Competing interests:** The authors have declared that no competing interests exist.

## Introduction

Backfilling in the stope plays a crucial role in providing support to the surrounding rock mass and improving its stress distribution, thereby ensuring a safe working environment for personnel within the stope [1, 2]. Hence, the mechanical properties and seepage characteristics of backfill have emerged as key concerns for engineers and technicians, as they are influenced by factors such as matrix materials, cement content, and the incorporation of reinforcement agents [3–6]. Pyrite and associated pyrite are common iron ore resources in China, resulting in the production of sulfur tailings [7]. During the surface stockpiling process, the stability of the tailings reservoir dam is threatened by sulfur tailings, which also has environmental implications [8]. Studies conducted by Yin et al. [9] indicate that the strength of CSTB decreases with increasing sulfur content. Chen et al. [10] suggest that sulfur content promotes early compressive strength but adversely affects later compressive strength of backfill. Dong et al. [11] found that sulfur content enhances early strength but inhibits later compressive strength. Chen et al. [12] propose that tailing sulfide adversely affects the setting time and compressive strength of backfill. Ercikdi et al. [13] conducted compressive strength tests on backfill after desulfurization of tailings and concluded that the strength of desulfurized backfill significantly improves.

Additionally, the addition of fiber is known to enhance strength and restrain deformation [14]. Xue et al. [15] studied the pressure resistance characteristics of fiber-reinforced tailings backfill and found that fiber-reinforced backfill exhibits the characteristic of "cracking but not breaking." Chen et al. [16] discovered that polypropylene fiber contributes to the strength of backfill and restrains the occurrence of cracks. Cao et al. [17] investigated the compressive strength and microstructure of fiber-reinforced backfill and noted that the addition of fiber improves the toughness of backfill. Wang et al. [18] examined the influence of rubber fiber on certain properties of backfill and observed that increasing rubber fiber content decreases the compressive strength but significantly improves the toughness of backfill. Xue et al. [19] studied the physical properties of fiber-reinforced backfill under triaxial compression and reported that the appropriate addition of fiber enhances the compressive strength of backfill and inhibits the development of failure cracks. Similarly, Xue et al. [20] investigated the physical properties of fiber-reinforced coarse aggregate cemented backfill and found that the appropriate addition of fiber improves the compressive strength and limits the development of failure cracks. Furthermore, Xue et al. [21] studied the properties of polypropylene fiber-modified coal-based solid waste filling materials and highlighted the effective enhancement of strength and toughness by polypropylene fiber. These studies primarily focused on the influence of matrix materials (such as sulfur tailings) and the addition of reinforcing agents, with subsequent analysis of the failure mechanisms.

However, as an artificially composite weakly cementing material, the energy consumption characteristics of cemented backfill during deformation and failure vary under different material and external load conditions. The yield failure and damage of cemented backfill are fundamentally processes of energy dissipation. Understanding the energy evolution characteristics of backfill has significant engineering value for the prevention and prediction of stope backfill disasters in practical applications [22]. In underground mining engineering, mining activities, disturbance, or pressure relief of the ore body are always accompanied by energy input, accumulation, dissipation, and release [23]. When backfill is used as a pillar or roof, the induced energy from mining the ore body and the migration of rock masses transfers energy into the backfill. Part of the energy accumulates as elastic strain energy within the backfill and is released upon damage, while the remaining energy is dissipated as electromagnetic radiation or acoustic emission [24]. Therefore, investigating the failure mechanism of backfill based on

energy dissipation may provide a more universal and closer understanding of the deformation and failure nature of backfill [25].

Researchers such as Xu et al. [26] have analyzed the effect of confining pressure on the energy dissipation and deformation failure of backfill through triaxial compression tests. Hou et al. [27, 28] systematically studied the influence of loading rate and curing age on the mechanical properties and energy dissipation of backfill through uniaxial compression tests. However, the aforementioned studies did not consider the influence of matrix materials and additional reinforcement agents on the energy dissipation of backfill, and there have been few reports on the influence of matrix materials and additional reinforcement agents on the energy dissipation characteristics of backfill. In the field of rock mechanics, related research has been conducted. For example, Zhao et al. [29] discovered that increasing loading rate significantly improves the elasticity of rock. Xie et al. [30] studied energy dissipation, transfer, and release from the perspective of non-equilibrium thermodynamics. Zhang et al. [31] found that the amplitude of pre-peak energy mutation decreases with increasing fracture length. However, as backfill and rock are two distinct media with substantial differences in strength, material properties, inhomogeneity, and meso-structure, the changes in matrix materials, added reinforcement agents, and curing age are crucial factors in energy evolution.

Therefore, this study focuses on CSTB as the research subject and analyzes the influence of polypropylene fiber content and length on the mechanical properties and energy evolution of CSTB through uniaxial compression tests. The innovative aspects of this study are twofold. Firstly, the mechanical property tests of backfill are employed to analyze the influence of polypropylene fiber content and length on the stress, toughness, and deformation characteristics of CSTB. Secondly, based on the principle of energy dissipation, the analysis examines the influence of polypropylene fiber content and length on characteristic parameters related to the energy of CSTB, thereby revealing the energy damage evolution mechanism of CSTB under uniaxial compression. To achieve these goals, uniaxial compression tests were conducted on CSTB, and the mechanical parameters were statistically analyzed. Additionally, energy parameters of the backfill were calculated using the principle of energy dissipation during indoor uniaxial compression, and the variation characteristics of these energy parameters were analyzed. Finally, scanning electron microscopy (SEM) was employed to investigate the microstructure of CSTB, revealing the mechanism by which fiber reinforcement enhances the mechanical properties of CSTB.

## Experimental materials and methods

### Experimental materials

Sulfur tailings with different sulfur content were prepared by mixing sulfur concentrate and common tailings. Fig 1 describes the particle size distribution of Test materials. Composite Portland cement (P.C.32.5) was used as the cementing material in this test. Tables 1 and 2 show the chemical composition of sulfur concentrate, common tailings and cement used in the test. The basic physical and mechanical properties of polypropylene fiber are shown in Table 3.

### Experimental scheme and preparation of CSTB specimens

According to the filling body parameters of a pyrite in Anhui province, the mass concentration of the slurry was 73% and cement-to-tailings (c/t) ratio is 1:8. The specific test scheme was shown in Table 4. Samples numbered A1~A4 are mainly used to analyze the influence of sulfur content on the physical properties of CSTB. In addition, samples numbered B1~B12 were mainly used to analyze the influence of fiber characteristic parameters (fiber length and

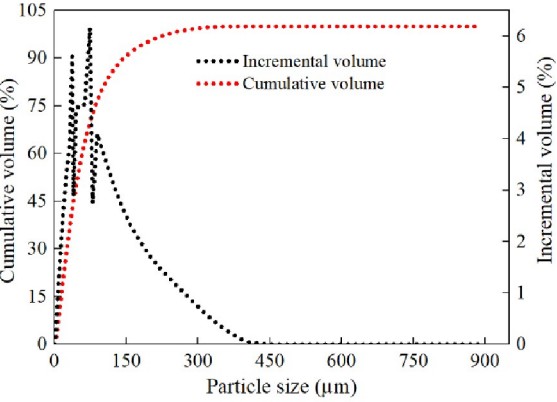

(a) sulfur concentrate

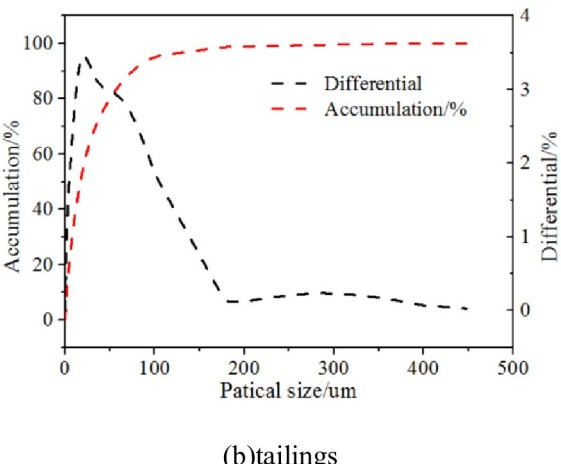

(b)tailings

**Fig 1. Particle size distribution of sulfur concentrate and tailings.** (a) sulfur concentrate and (b) tailings.

dosage) on the mechanical properties of CSTB. In this experiment, the fiber length is mainly 3mm, 6mm and 12mm. Combined with relevant research results, when the fiber content is 0.25%-0.75%, the fiber has the best improvement effect on the physical properties of CTB. Therefore, 0.2%, 0.4%, 0.6% and 0.8% fiber content were chosen [32].

According to the test plan in Table 4, sulfur concentrate and ordinary tailings were mixed to produce sulfur tailings with sulfur content of 6%, 12%, 18% and 25%. In addition, fiber, cement and sulfur tailings were mixed and stirred according to the test plan, and then water was added to prepare fiber reinforced sulfur tailings filling slurry. It should be noted that the added fiber uses the "dry mixing method", that is, in the dry material mixing process, the fiber is divided into a small number of times into the dry material, to ensure that the fiber is fully

**Table 1. Chemical composition of sulfur concentrate (mass fraction) %.**

| Ingredient | C | O | Si | Ca | Fe | S | Cu | Mg | others |
|---|---|---|---|---|---|---|---|---|---|
| sulfur concentrate | 1.07 | 26.70 | 2.86 | 2.09 | 35.8 | 27.0 | 0.35 | 1.22 | 2.91 |

**Table 2. Chemical composition of cement and tailings (mass fraction) %.**

| Ingredient | $SiO_2$ | CaO | MgO | $Al_2O_3$ | $Fe_2O_3$ | $K_2O$ | $TiO_2$ | MnO | others |
|---|---|---|---|---|---|---|---|---|---|
| Tailings | 42.20 | 3.73 | 32.71 | 4.04 | 12.14 | 0.39 | 0.33 | 0 | 4.22 |
| cement | 28.36 | 48.28 | 2.50 | 11.87 | 2.88 | 1.07 | 0.60 | 0.12 | 4.31 |

dispersed after adding water and stirring for 3 to 5 minutes to prepare an evenly mixed filling slurry.

Finally, a cylinder model (diameter: 50mm, height: 100mm) was used to prepare the CSTB sample. Finally, the sample was placed in the standard curing box for curing, and the sample was loaded after the curing time reached 28d. Fig 2 shows polypropylene fibers of different lengths. The fiber lengths in Fig 2 are 3mm, 6mm and 12mm respectively. Due to the large length span of the fiber, the influence of the fiber length on the mechanical properties of the filling can be well analyzed. Fig 3 shows test raw materials and CSTB samples.

## Testing methods

**Uniaxial compression and tensile strength testing.** The mechanical strength test mainly includes compressive strength and tensile strength test. Wdw-50 electronic universal testing machine was used as loading equipment to conduct mechanical strength test on the backfill. The test equipment is shown in Fig 4.

**Scanning electron microscope test.** SEM was used to test the microstructure of CSTB, so as to reveal the influence mechanism of fiber on CSTB. The electron microscope model is JSM-6510A. The resolution and maximum acceleration voltage are 6 nm and 30 kV, respectively. The test equipment is shown in Fig 5.

**Principle of energy dissipation.** There are energy input, accumulation, dissipation and energy transformation in deformation. It is assumed that the energy released by heat radiation and heat exchange is ignored. Therefore, the calculation relation of CSTB is as follows [33]

$$U = U^e + U^d \tag{1}$$

In the above formula: $U$ is the total input energy, and $U^d$ is the dissipated energy, which is used to form internal damage and plastic deformation of CSTB; $U^e$ is the elastic strain energy, whose energy density is $kJ/m^3$.

Fig 6 shows the relationship between elastic strain energy and dissipation energy of CSTB during deformation and failure.

In the principal stress space, the $U$ and $U^e$ in the backfill element can be expressed as:

$$U = \int_0^{\varepsilon_1} \sigma_1 d\varepsilon_1 + \int_0^{\varepsilon_2} \sigma_2 d\varepsilon_2 + \int_0^{\varepsilon_3} \sigma_3 d\varepsilon_3 \tag{2}$$

$$U^e = \frac{1}{2}\sigma_1\varepsilon_1^e + \frac{1}{2}\sigma_2\varepsilon_2^e + \frac{1}{2}\sigma_3\varepsilon_3^e \tag{3}$$

By substituting Hooke's theorem into Formula (3), the elastic strain energy can be simplified as:

$$U^e = \frac{1}{2E}\left[\sigma_1^2 + \sigma_2^2 + \sigma_3^2 - 2\mu(\sigma_1\sigma_2 + \sigma_1\sigma_3 + \sigma_3\sigma_2)\right] \tag{4}$$

**Table 3. Physical and mechanical parameters of polypropylene fiber.**

| Length/mm | Diameter/mm | Tensile strength/MPa | Elastic modulus/GPa | Density/$(g/m^3)$ | Elongation /% | Elongation rate/% |
|---|---|---|---|---|---|---|
| 3, 6, 12 | 19 | 350 | 3.5 | 910 | 30 | 28 |

**Table 4. Designed schedule for the experiments.**

| Serial number | Mass concentration /% | Cement-sand ratio | sulphidic tailings | Sulfur content /% | Fiber length /mm | Fiber content /% |
|---|---|---|---|---|---|---|
| A1 | 73 | 1:8 | T1 | 6 | / | / |
| A2 | 73 | 1:8 | T2 | 12 | / | / |
| A3 | 73 | 1:8 | T3 | 18 | / | / |
| A4 | 73 | 1:8 | T4 | 25 | / | / |
| B1 | 73 | 1:8 | T3 | 18 | 3 | 0.2 |
| B2 | 73 | 1:8 | T3 | 18 | | 0.4 |
| B3 | 73 | 1:8 | T3 | 18 | | 0.6 |
| B4 | 73 | 1:8 | T3 | 18 | | 0.8 |
| B5 | 73 | 1:8 | T3 | 18 | 6 | 0.2 |
| B6 | 73 | 1:8 | T3 | 18 | | 0.4 |
| B7 | 73 | 1:8 | T3 | 18 | | 0.6 |
| B8 | 73 | 1:8 | T3 | 18 | | 0.8 |
| B9 | 73 | 1:8 | T3 | 18 | 12 | 0.2 |
| B10 | 73 | 1:8 | T3 | 18 | | 0.4 |
| B11 | 73 | 1:8 | T3 | 18 | | 0.6 |
| B12 | 73 | 1:8 | T3 | 18 | | 0.8 |

When calculating the $U^e$, the initial elastic modulus $E_0$ is used to replace the unloading elastic modulus $E$ [34]. For the uniaxial compression test of backfill, Formula (4) can be further simplified as:

$$U^e = \frac{1}{2E_0}\sigma_1^2 \tag{5}$$

The total energy is calculated by using the basic concept of calculus and the sum of the areas of tiny rectangles. The calculation formula is as follows:

$$U = \sum_{i=1}^{n}(\sigma_1^i + \sigma_1^{i+1})(\varepsilon_1^{i+1} - \varepsilon_1^i) \tag{6}$$

In the above formula: $\sigma_1^i$ and $\varepsilon_1^i$ are the axial stress value and axial strain value of stress-strain curve respectively. Therefore, the dissipated energy of CSTB can be calculated by Eq (7):

$$U^d = U - U^e \tag{7}$$

## Results and analysis

### Stress-strain curve characteristics and failure stage division of CSTB

Fig 7(A) shows the stress-strain curves of CSTB under uniaxial loading with different sulfur contents. Fig 7(B) and 7(C) show the stress-strain curve characteristics of CSTB with 3mm and 12mm fibers respectively. Fig 7(D) shows the typical stress-strain curve of CSTB. It can be seen from Fig 7 that the stress-strain curve of the CSTB also changes to a certain extent when polypropylene fiber is added. The stress-strain curve of the backfill without fiber decreases very suddenly when the load on the CSTB exceeds the peak stress, showing obvious brittle failure characteristics [35]. However, the stress-strain curve of the fiber reinforced CSTB does not decrease rapidly when the load exceeds the stress peak, indicating that the fiber addition improves the ability of the CSTB to resist deformation and failure [36].

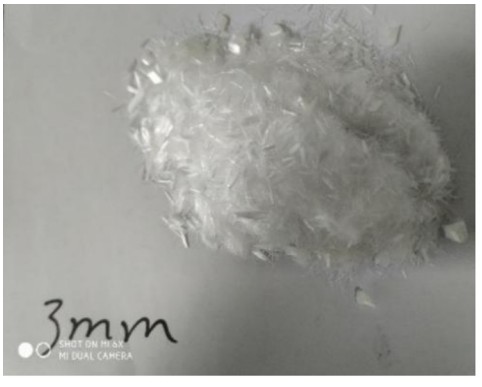 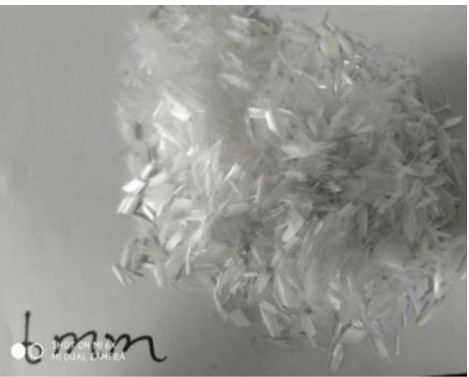

(a) 3 mm length                                        (b) 6 mm length

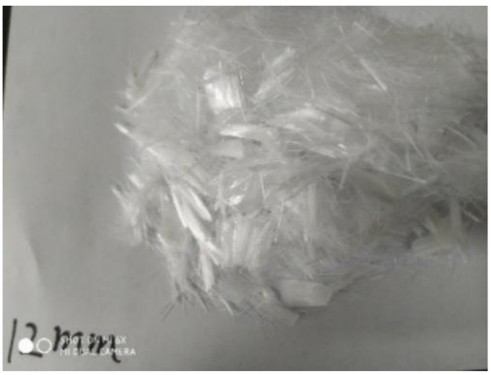

(c) 12 mm length

**Fig 2. Polypropylene fibre.** (a) 3 mm length (b) 6 mm length, and (c) 12 mm length.

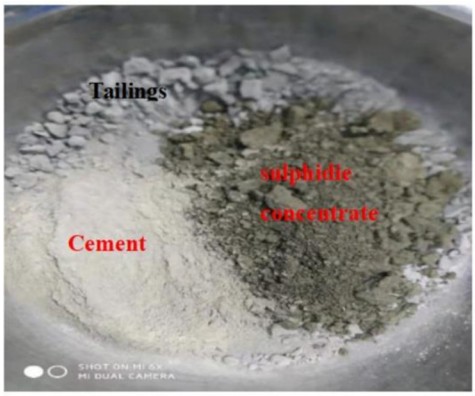 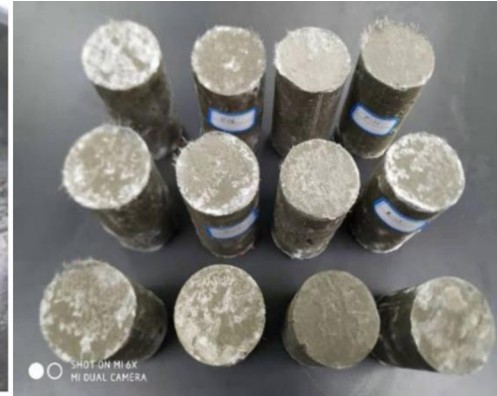

(a) experimental materials                        (b) some samples

**Fig 3. Experimental materials and some samples.** (a) experimental materials and (b) some samples.

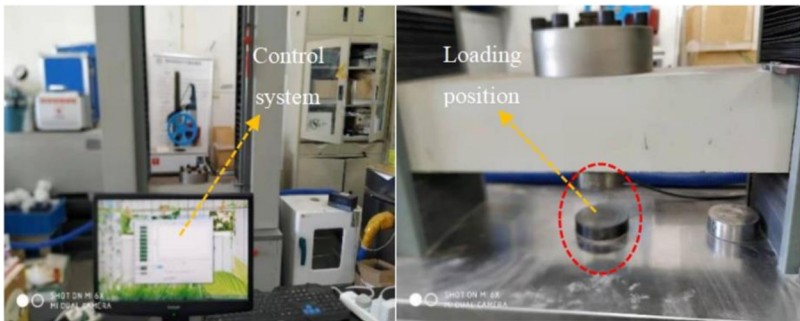

**Fig 4. Experimental loading equipment.**

In addition, from Fig 7(D), the whole deformation is four stages: (1) Initial fracture compaction stage (OA). At this stage, the stress value shows A trend of slowly increasing, and when the stress value reaches point A, it means that the initial pores inside the sample are compressed. (2) Linear elastic deformation stage (AB). When the stress value reaches point B, new cracks start to occur inside the CSTB, which means that the CSTB will enter the stage of yield failure. The stress value of CSTB at point B is called initial fracture stress value $\sigma_1$, which is generally 75% peak stress [36]. (3) Yield failure stage (BC). The stress value of CSTB at point C is the peak stress $\sigma_2$. When the stress value exceeds the peak stress, the CSTB enters the post-peak failure stage. (4) Post-peak failure stage (CD). The existence of residual stress $\sigma_3$ indicates that the CSTB still has a certain bearing capacity after failure.

Fig 8 shows the relationship between peak stress, residual stress and fiber content of CSTB. As can be seen from Fig 8, when the fiber content is between 0% and 0.6%, the fiber content is

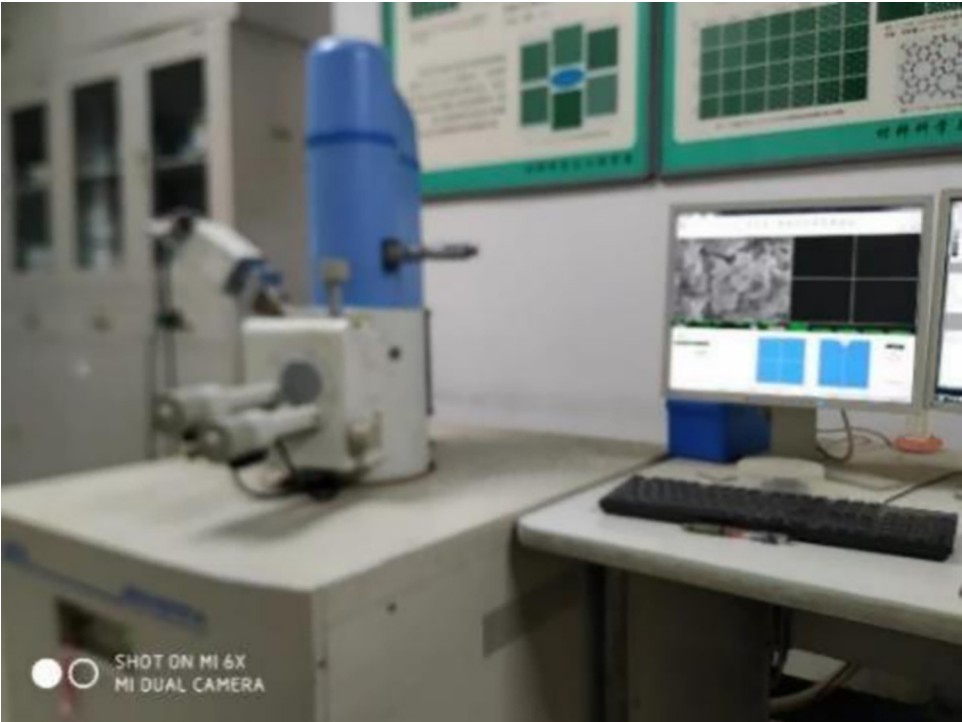

**Fig 5. Scanning electron microscope (SEM).**

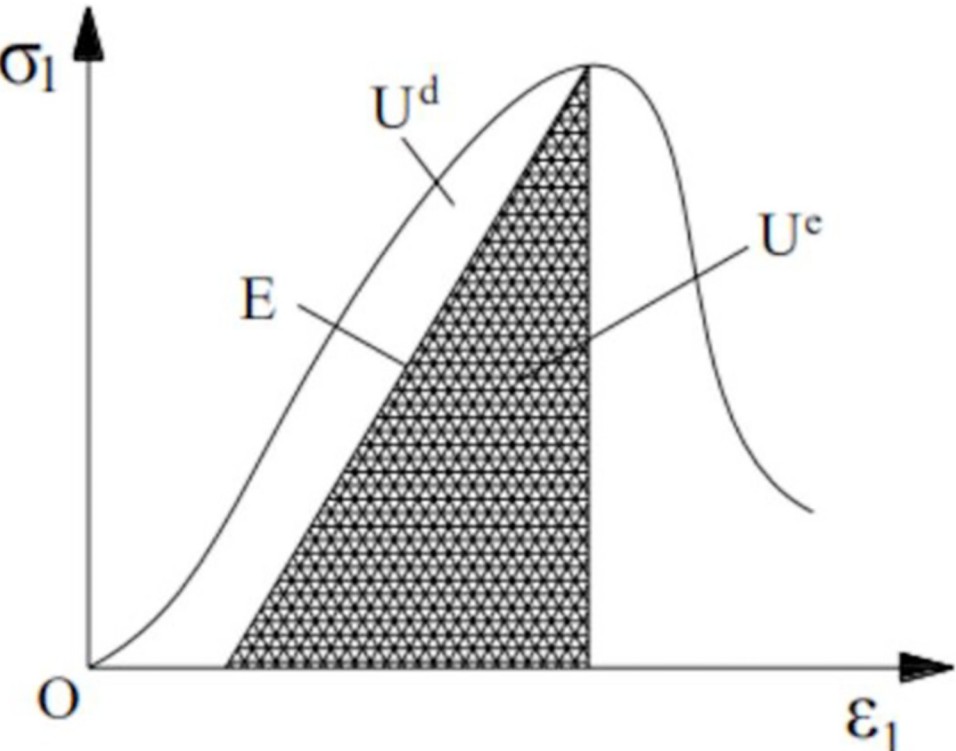

**Fig 6. Relationship between elastic strain energy and dissipated energy of CSTB during deformation and failure.**

positively correlated with the stress value of backfill. However, when the fiber content is between 0.6% and 0.8%, the fiber content and the stress value of backfill are negative correlation. Therefore, the 0.6% fiber content is the optimal. The fiber content goes from 0% to 0.6%, and the fiber length is 3mm, 6mm and 12mm, the peak stress increases by 31.8%, 49.7% and 30.9%. For residual stress, it increases by 474%, 509% and 472%. Therefore, fiber can not only increase the peak stress, but also significantly increase the residual stress of CSTB. From the variation of residual stress, it shows that fiber can not only effectively improve the mechanical properties of CSTB, but also greatly improve the bearing capacity of CSTB. Moreover, the maximum increasment of the stress value is when the fiber length is 6mm, indicating that the polypropylene fiber length of 6mm is optimum at 0.6% fiber content. In addition, the relevant results obtained in this study are compared with the existing results to illustrate the necessity and reliability of the research results in this paper. Table 5 shows the comparison between the results of this paper and other existing results of the same type. As can be seen from Table 5, when sulfur tailings or ordinary tailings are used to prepare backfill, adding appropriate amount of fiber can improve the mechanical properties of both types of backfill, but the improvement effect is significantly related to the type and dosage of fiber. At the same time, straw fiber can also improve the mechanical properties of cemented sulfur tailings backfill, but the improvement effect is lower than that of polypropylene fiber on the physical properties of cemented sulfur tailings backfill.

## Effect of polypropylene fiber on toughness properties of CSTB

In the goaf, the CSTB not only have good compressive strength, but also have good toughness to ensure that the CSTB has good anti-deformation and failure ability [40]. In this study, peak

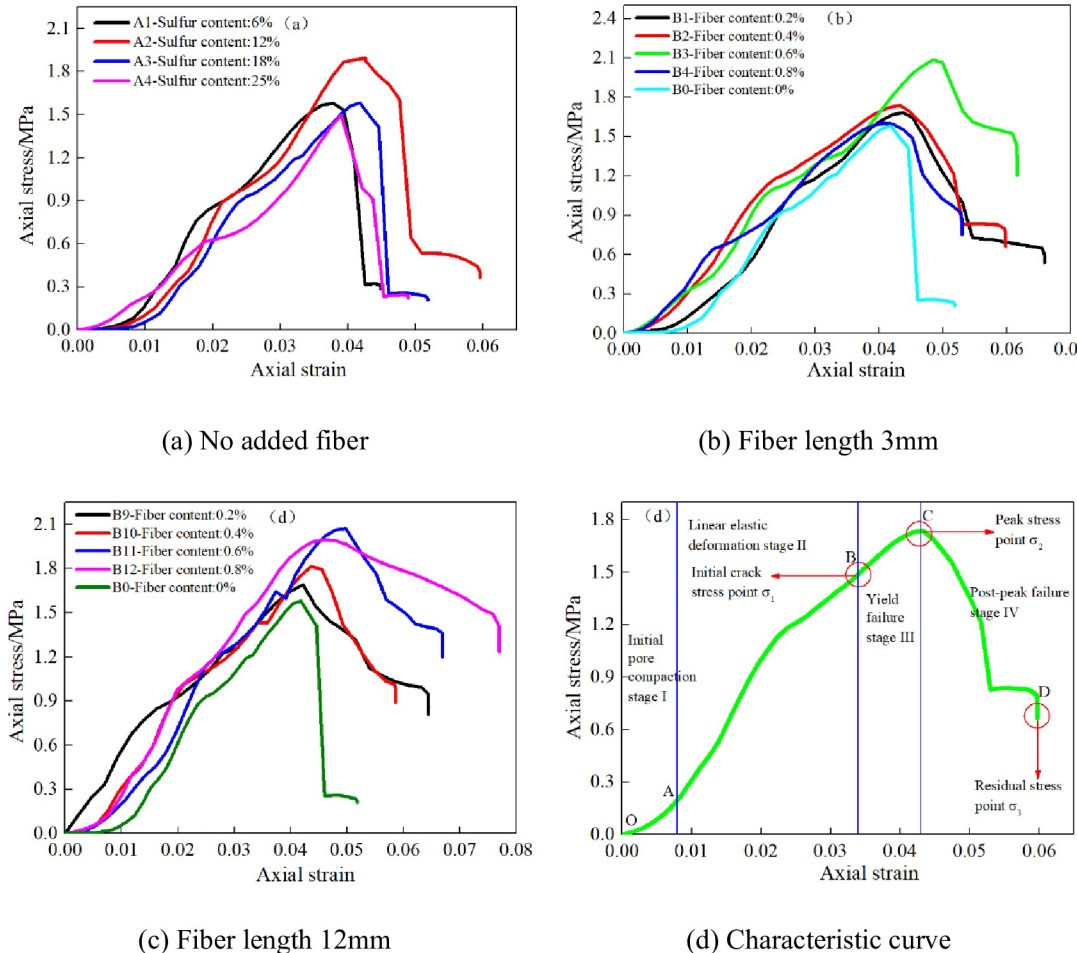

**Fig 7. Stress-strain curve of backfill under uniaxial loading.** (a) No added fiber, (b) Fiber length 3mm, (c) Fiber length 12mm, and (d) Characteristic curve.

strain factor K was used to define toughness evaluation parameter [41]. Therefore, peak strain factor K can be calculated by Formula (8).

$$K = \frac{\varepsilon_1}{\varepsilon_2} \tag{8}$$

Where $\varepsilon_1$ represents the peak strain of fiber reinforced CSTB; $\varepsilon_2$ represents the peak strain of CSTB without fiber.

Fig 9 shows the toughness curve of fiber reinforced CSTB. In Fig 9, the peak strain factor K of the fiber reinforced CSTB increases at first, and then decreases, and reached the maximum value when the fiber content was 0.6%. For polypropylene fiber with a fiber length of 3mm, the peak strain factor K of the fiber-reinforced CSTB is 1.045, 1.033, 1.158 and 0.971 when the fiber content is 0.2%, 0.4%, 0.6%, and 0.8%. For polypropylene fiber with a fiber length of 6mm, the peak strain factor K of the fiber-reinforced CSTB is 1.036, 1.074, 1.211 and 1.119 when the fiber content is 0.2%, 0.4%, 0.6%, and 0.8%. For polypropylene fiber with a fiber length of 12mm, the peak strain factor K of the fiber-reinforced CSTB is 1.009, 1.043, 1.191 and 1.098 when the fiber content is 0.2%, 0.4%, 0.6%, and 0.8%. Therefore, the change of fiber length will also have a certain influence on the peak strain factor K of the CSTB. When the

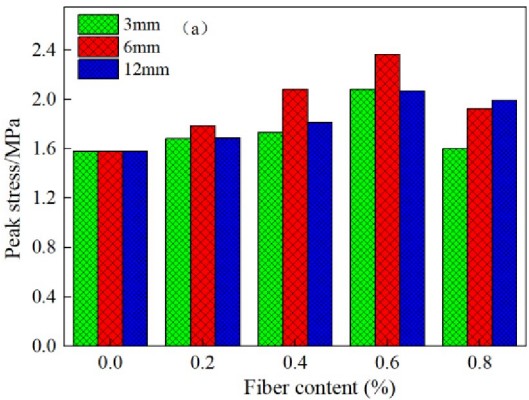

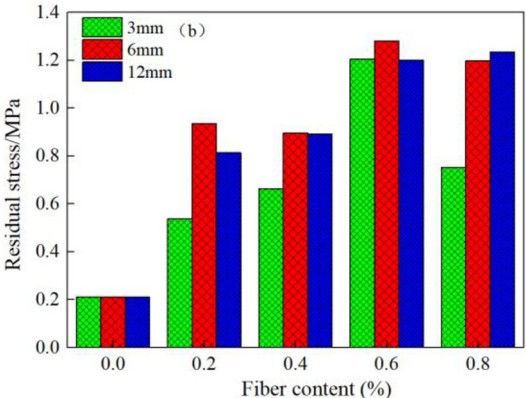

**Fig 8. Relationship between peak stress, residual stress and fiber content of fiber reinforced CSTB.**

fiber content is 0.6%, the influence degree of fiber length on the peak strain factor K of CSTB is 6mm>12mm>3mm. Fig 9 shows the functional relationship between peak strain factor K of CSTB and fiber content at different fiber lengths.

### Effect of fiber on tension-compression ratio of CSTB

The tension-compression ratio is the ratio of the tensile strength to the compressive strength. The deformation performance and crack resistance of CSTB are positively correlated with the tension-compression ratio [42]. The tension-compression ratio of fiber reinforced CSTB is shown in Fig 10. Fig 10 shows that the tension-compression ratio of the CSTB with

**Table 5. Comparison with existing research results.**

| Relevant achievement | Fibre type | Tailings type | Fiber content /% | Fibre length /mm | Peak stress /MPa | Amplitude of stress increase |
|---|---|---|---|---|---|---|
| Xu et al. [37] | Polypropylene fibre | Plain tailings | 0.25 | 6 | 2.31 | 29.41% |
| Ruan et al. [38] | Straw fiber | Sulphur-bearing tailings | 0.6 | 2~5 | 3.23 | 14.29% |
| Zhao et al. [39] | Glass fiber | Plain tailings | 0.5 | 6 | 1.48 | 15.60% |
| Achievements of this paper | Polypropylene fibre | Sulphur-bearing tailings | 0.6 | 6 | 2.38 | 50.60% |

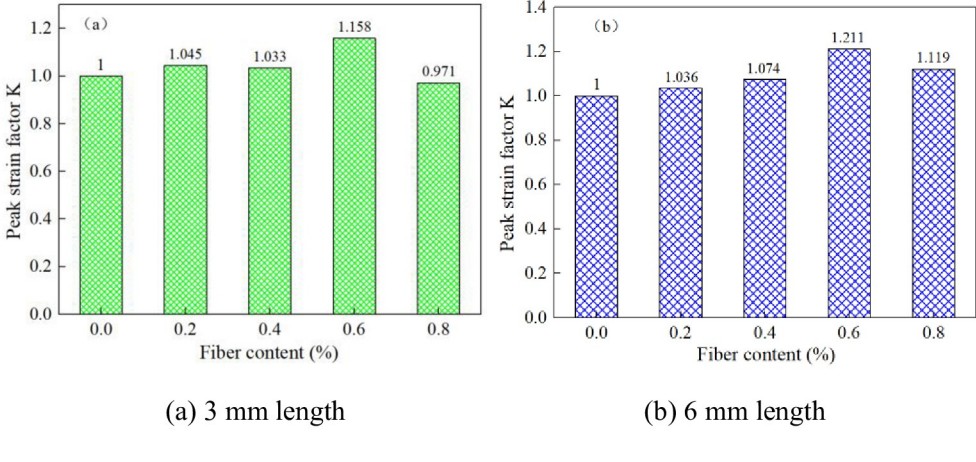

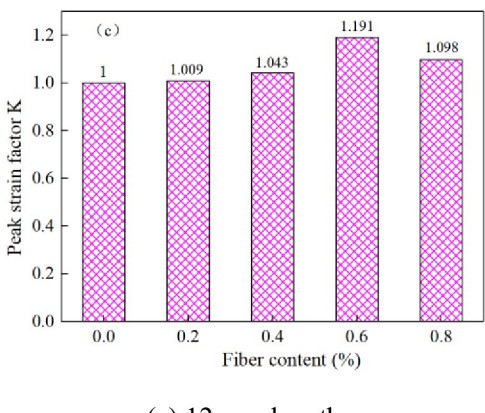

**Fig 9. Toughness curves of fiber reinforced CSTB.** (a) 3 mm length, (b) 6 mm length, and (c) 12 mm length.

polypropylene fiber is more bigger compared with that of no-fiber. It illustrate that fiber can enhance the deformation and crack resistance. The tension-compression ratio of the FR-CSTB with fiber content of 0.2%, 0.4%, 0.6% and 0.8% are 0.195, 0.206, 0.215 and 0.229 when the fiber length is 3mm. The tension-compression ratio of the fiber reinforced CSTB with fiber content of 0.2%, 0.4%, 0.6% and 0.8% are 0.226, 0.201, 0.194 and 0.219 when the fiber length is 6mm. The tension-compression ratio of the fiber reinforced CSTB with fiber content of 0.2%, 0.4%, 0.6% and 0.8% are 0.237, 0.234, 0.218 and 0.195 when the fiber length is 12mm. Therefore, when the fiber length is 3mm, the tension-compression ratio of the CSTB increases as fiber content increasing and reaches the maximum value when the fiber content is 0.8%. However, when the fiber length was 12mm, the tension-compression ratio decreases with the increase of fiber content. Moreover, it reached the maximum value at 0.2%. The fibers with longer length will intertwine and become cohesive, leading to the stress concentration phenomenon of the CSTB under load as fiber content increasing [41]. Therefore, when the fiber length is 12mm, because the fiber length is too long, the fibers contact each other to form a stress concentration point, which limits the enhancement effect of the fiber against tensile strength, resulting in the tensile compression ratio decreases with the increase of the fiber content [41]. At the same time, when the fiber content was fixed, the tension-compression ratio of

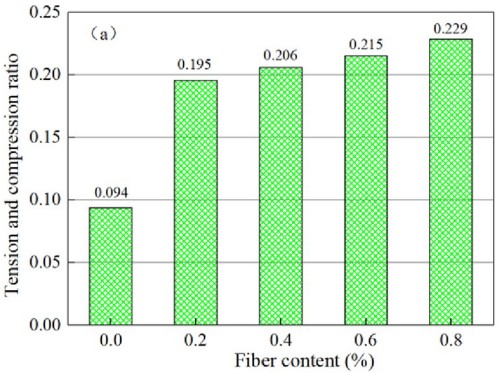

(a) 3 mm length

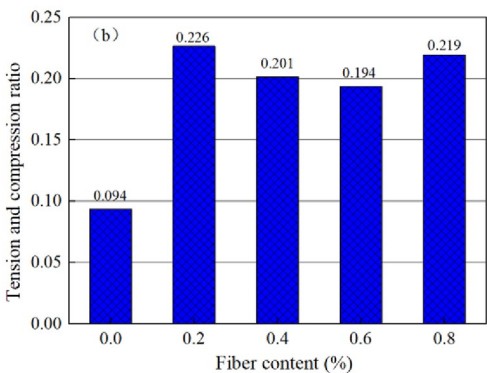

(b) 6 mm length

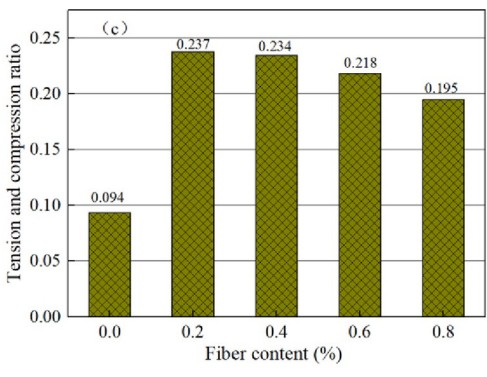

(c) 12 mm length

**Fig 10. Tension and compression ratio of CSTB.** (a) 3 mm length, (b) 6 mm length, and (c) 12 mm length.

the CSTB with fiber length of 12mm was significantly higher than that with polypropylene fiber length of 3mm, indicating that the increase of fiber length could also improve the deformation and crack resistance of the CSTB. The reason why fiber length increasing can improve the deformation and crack resistance of CSTB is that the cohesion between fiber and cement-tailings matrix makes the increase of fiber length conducive to the formation of a wider force transfer system. Therefore, when the CSTB is loaded, the sample can uniformly bear the external load, so as to improve the deformation and crack resistance of the CSTB.

### Energy damage evolution mechanism of CSTB

**Energy parameters of CSTB at characteristic stress point.** The energy characteristic parameters of CSTB in different deformation and failure stages are different. Table 6 shows the characteristic energy parameters of CSTB at peak stress $\sigma_2$. Meanwhile, the ($U^d/U^e$) of the CSTB was defined as the correction coefficient of brittleness index BIM, and BIM value was positively correlated with the ductile plastic performance of the CSTB [46]. In Table 6, at the peak stress, the trend of dissipated energy, total strain energy, elastic strain energy is consistent with the above as fiber content increasing. Also, the maximum of them is 0.6% fiber content. Elastic strain energy of CSTB at the peak stress point can be regarded as the energy storage limit of CSTB [43]. When the fiber length is 3mm, the energy storage limit of CSTB increases by 5.8%, 17.5%, 40.8% and 14.1% as fiber content going 0% to 0.8%. When the fiber length is 6mm, the energy storage limit of CSTB increases by 16.4%, 44.9%, 77.8% and 48.9%. When the fiber length is 12mm, the energy storage limit of CSTB increases by 16.7%, 26.7%, 55.3% and 44.9%. Therefore, the addition of polypropylene fiber is beneficial to the energy storage, so that the linear elastic deformation stage of the CSTB can be extended to a higher level, and the peak stress of the backfill can be increased macroscopically. At the same time, The increase of fiber length is also conducive to improving the energy storage limit of CSTB as fiber content remain unchanged. In addition, the CSTB with fiber has a higher BIM value, indicating that fiber is important for improving the ductility of the filling body, thus improving its crack resistance.

**Table 6. Energy characteristic parameters of CSTB at peak stress point under uniaxial loading.**

| Fiber length | Fiber content/% | Peak stress point / (kJ/m$^3$) | | | Energy parameter ratio | | |
|---|---|---|---|---|---|---|---|
| | | $U$ | $U^e$ | $U^d$ | ($U^e/U$)/% | ($U^d/U$)/% | BIM |
| 3mm | 0 | 27.999 | 24.986 | 3.013 | 89.2 | 10.8 | 0.121 |
| | 0.2 | 32.937 | 26.434 | 6.502 | 80.3 | 19.7 | 0.246 |
| | 0.4 | 40.345 | 29.445 | 10.900 | 73.0 | 27.0 | 0.370 |
| | 0.6 | 48.508 | 35.204 | 13.304 | 72.5 | 17.5 | 0.379 |
| | 0.8 | 33.302 | 28.505 | 4.796 | 85.6 | 14.4 | 0.168 |
| 6mm | 0 | 27.999 | 24.986 | 3.013 | 89.2 | 10.8 | 0.121 |
| | 0.2 | 39.517 | 29.083 | 10.074 | 73.6 | 26.4 | 0.346 |
| | 0.4 | 47.276 | 36.218 | 11.058 | 76.6 | 23.4 | 0.305 |
| | 0.6 | 56.644 | 44.437 | 12.207 | 78.4 | 21.6 | 0.275 |
| | 0.8 | 47.507 | 37.216 | 10.291 | 78.3 | 21.7 | 0.276 |
| 12mm | 0 | 27.999 | 24.986 | 3.013 | 89.2 | 10.8 | 0.121 |
| | 0.2 | 39.728 | 29.165 | 10.563 | 73.4 | 26.6 | 0.362 |
| | 0.4 | 42.495 | 31.653 | 11.245 | 74.5 | 25.5 | 0.355 |
| | 0.6 | 49.544 | 38.299 | 11.245 | 77.3 | 22.7 | 0.293 |
| | 0.8 | 46.043 | 36.193 | 9.849 | 78.6 | 21.4 | 0.272 |

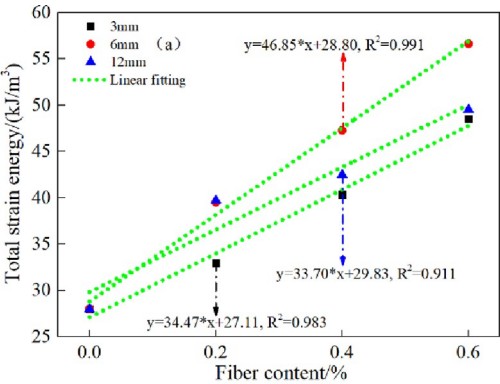

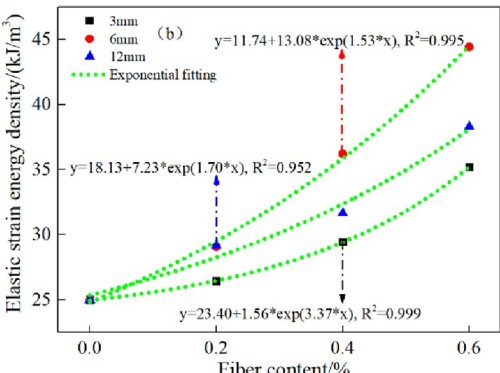

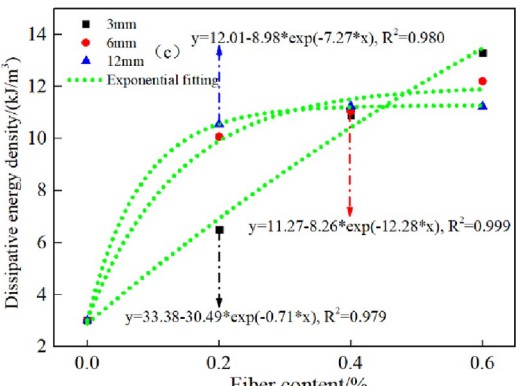

**Fig 11. Energy variation curve of CSTB at peak stress point under uniaxial loading.**

Fig 11 shows the relationship between energy characteristic parameters of CSTB and fiber content. It shows that the total strain energy of CSTB basically follows the law of linear function increase with the increase of fiber content, and the total strain energy of CSTB with 6mm polypropylene fiber increases fastest. The elastic and dissipated energy both increase exponentially as fiber content increasing. In addition, the dissipated energy increases as fiber content increasing, indicating that the development of cracks inside the CSTB needs to consume more energy, which reflects that the fiber incorporation improves the yield strength of the CSTB.

Table 7. Energy characteristic parameters of backfill at residual stress point.

| Fiber length | Fiber content /% | Peak stress point / (kJ/m$^3$) | | | Energy parameter ratio | |
|---|---|---|---|---|---|---|
| | | $U$ | $U^e$ | $U^d$ | $U^e/U$ | $U^d/U$ |
| 3mm | 0 | 34.836 | 0.441 | 34.396 | 7.0 | 93.0 |
| | 0.2 | 54.814 | 2.696 | 52.117 | 4.9 | 95.1 |
| | 0.4 | 60.099 | 4.287 | 55.811 | 7.1 | 92.9 |
| | 0.6 | 70.934 | 11.781 | 59.153 | 16.6 | 83.4 |
| | 0.8 | 49.085 | 6.301 | 42.783 | 12.8 | 87.2 |
| 6mm | 0 | 34.836 | 0.441 | 34.396 | 7.0 | 93.0 |
| | 0.2 | 66.266 | 7.956 | 58.310 | 12.0 | 88.0 |
| | 0.4 | 68.236 | 6.711 | 61.526 | 9.8 | 81.2 |
| | 0.6 | 84.436 | 12.980 | 71.456 | 15.4 | 84.6 |
| | 0.8 | 70.156 | 14.374 | 55.782 | 20.5 | 79.5 |
| 12mm | 0 | 34.836 | 0.441 | 34.396 | 7.0 | 93.0 |
| | 0.2 | 67.223 | 6.765 | 64.457 | 10.1 | 89.9 |
| | 0.4 | 65.135 | 7.617 | 57.518 | 11.7 | 88.3 |
| | 0.6 | 77.413 | 12.872 | 65.540 | 16.6 | 83.4 |
| | 0.8 | 70.645 | 11.861 | 58.784 | 16.8 | 83.2 |

Energy characteristic parameters of CSTB in the post-peak failure stage can reflect the capacity of CSTB to continue bearing loads [43]. Table 7 shows the energy characteristic parameters of CSTB at residual stress. When the fiber length is 3mm, the elastic strain energy of CSTB accounts for 4.9%, 7.1%, 16.6% and 12.8% of the total strain energy, while the dissipated energy accounts for 95.1%, 92.9%, 83.4% and 87.2% of the total strain energy. Therefore, the dissipated energy of the CSTB at the residual stress point is significantly greater than the elastic strain energy, which illustrates that the total energy input into the CSTB at the post-peak failure stage is mainly released as dissipated energy and the CSTB can basically no longer store the elastic strain energy.

## Energy distribution evolution law of CSTB deformation and failure

This paper takes the CSTB with 6mm polypropylene fiber as an example to analyze the energy distribution evolution characteristics of the CSTB with different fiber content, and reveals the internal relationship between energy value and axial strain. Fig 12 reflects the energy distribution evolution characteristics of CSTB with 6mm length fiber.

Fig 12 shows that under uniaxial loading, the energy CSTB is nonlinear as axial strain increasing. Meanwhile, the whole loading process can be summarized as:

1. Initial fracture compaction stage (OA). The energy value of CSTB increases slowly, and dissipated energy take up about 80%, 85%, 88% and 84% when the fiber content is 0.2%, 0.4%, 0.6% and 0.8%. Therefore, most of energy is consumed.

2. Linear elastic deformation stage (AB). At this stage, the elastic strain energy increases almost linearly, while the dissipation energy remains almost constant. At 0.2%, 0.4%, 0.6% and 0.8% fiber content, the elastic strain energy accounts for about 81%, 84%, 89% and 86%. Therefore, the energy is recoverable elastic energy at this stage. In addition, the dissipated energy of the CSTB is less than 0 at this stage, because the elastic modulus which is adopted to calculate the energy include the process of compaction [44, 45].

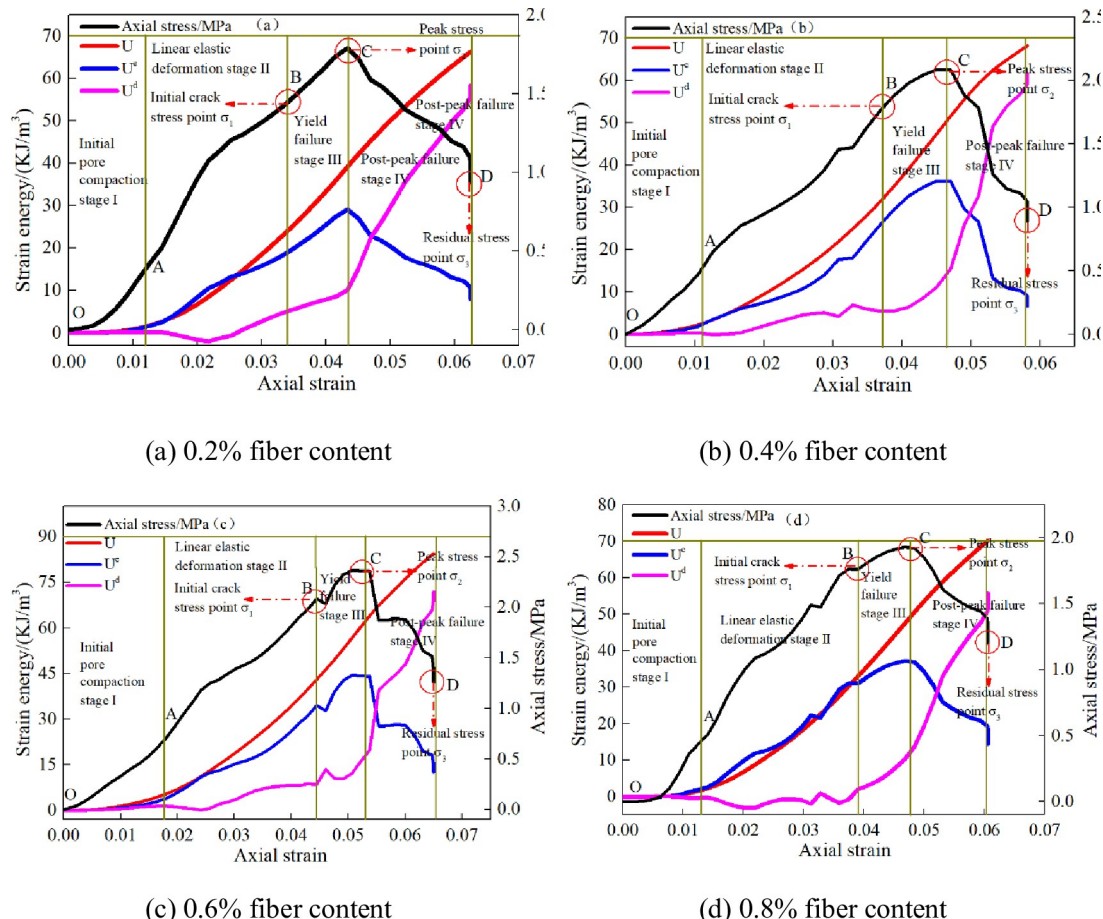

(a) 0.2% fiber content    (b) 0.4% fiber content

(c) 0.6% fiber content    (d) 0.8% fiber content

**Fig 12. Energy distribution evolution characteristics of CSTB.** (a) 0.2% fiber content, (b) 0.4% fiber content, (c) 0.6% fiber content, and (d) 0.8% fiber content.

3. Yield failure stage (BC). The increasement rate of elastic energy is gradually increased, and new pores and fissures generated inside the CSTB lead to the rapid growth of dissipative energy, but the proportion of elastic strain energy is still greater than that of dissipative energy. At 0.2%, 0.4%, 0.6% and 0.8% fiber content, the elastic strain energy accounts for about 73.6%, 76.6%, 78.4% and 78.3%, which illustrate that elastic strain energy also come into prominence.

4. Post-peak failure stage (CD). The elastic strain energy of CSTB decreases rapidly, while the dissipation energy shows an approximate linear increase trend. When the fiber content is 0.2%, 0.4%, 0.6% and 0.8%, dissipated energy take up 88.0%, 81.2%, 84.6% and 79.5%, indicating that the energy in the post-peak failure stage is mainly dissipated.

Under different fiber content, the energy parameters of CSTB show the same evolution law as axial strain increasing, which fiber content don't change the energy conversion process of CSTB, but only affects the horizontal range of energy value. The evolution curve of total strain energy shows increasing variation characteristics. The elastic energy change is form of "gentle—linear growth—slow growth—fast decline". Meanwhile, the dissipation energy curve is form of "gentle—slow growth—fast growth—linear growth".

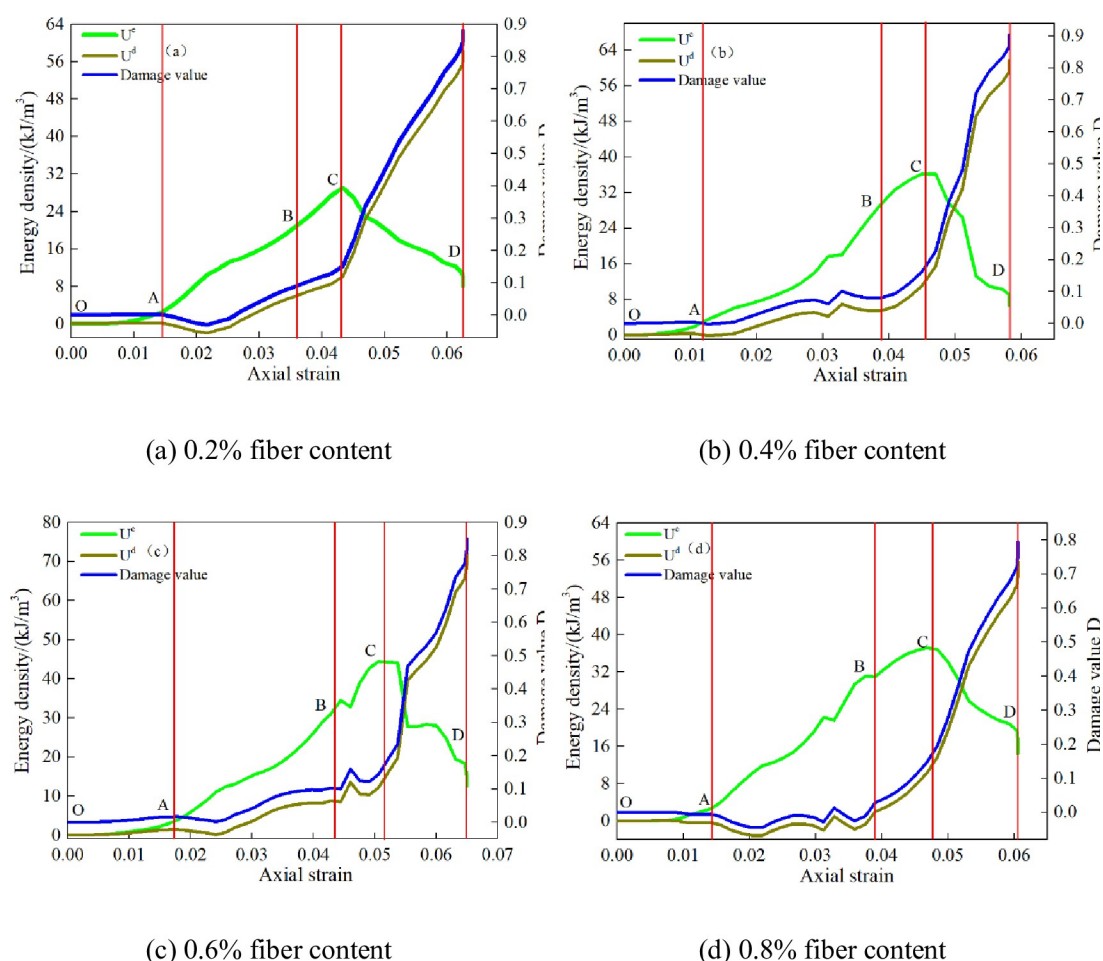

(a) 0.2% fiber content (b) 0.4% fiber content

(c) 0.6% fiber content (d) 0.8% fiber content

**Fig 13. Energy damage evolution curves of CSTB with fiber content.** (a) 0.2% fiber content, (b) 0.4% fiber content, (c) 0.6% fiber content, and (d) 0.8% fiber content.

### Energy damage evolution process of CSTB under uniaxial loading

The energy dissipation of CSTB is closely related to its strength attenuation, and the amount of dissipated energy can be used to reflect the damage degree of CSTB in the process of loading. Therefore, the ratio of dissipated energy to total input strain energy is assumed damage factor [46], there is:

$$D = \frac{U^d}{U} \tag{9}$$

Combined with Eqs (1) and (2), deriving:

$$D = \frac{\int_0^{\varepsilon_1} \sigma_1 d\varepsilon_1 - \frac{1}{2E_0}\sigma_1^2}{\int_0^{\varepsilon_1} \sigma_1 d\varepsilon_1} \tag{10}$$

The 6mm length of fiber was used for calculating (Fig 13). Therefore, energy consumption value and axial strain of CSTB, the energy damage evolution process of compressive failure of CSTB can be divided into four stages:

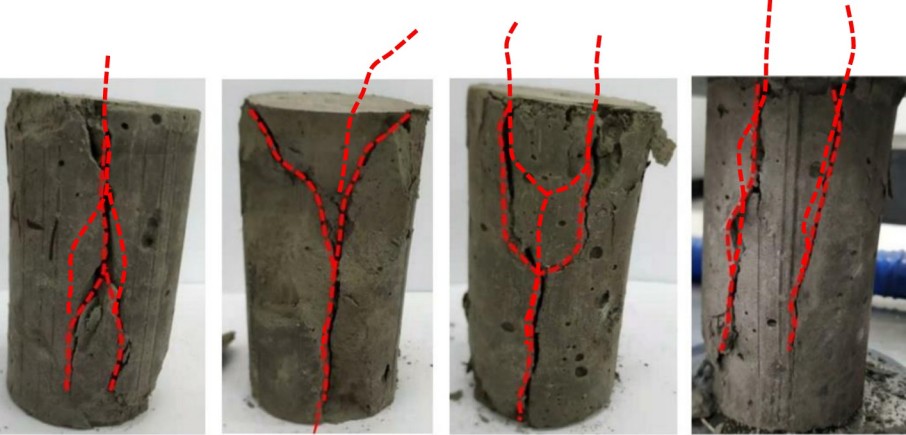

(a) 6% sulfur content    (b) 12% sulfur content    (c) 18% sulfur content    (d) 25% sulfur content

**Fig 14. Failure characteristics of CSTB with different sulfur contents.** (a) 6% sulfur content, (b) 12% sulfur content, (c) 18% sulfur content, and (d) 25% sulfur content.

1. Initial damage stage (OA), the initial compaction stage of stress-strain curve. In this stage, the damage value, dissipated energy and elastic strain energy of CSTB are small, and the damage value of CSTB can be ignored basically.

2. Slow growth stage of damage (AB), elastic deformation stage of stress-strain curve. At this stage, the damage value D shows a slow growth trend, and the existence of the damage value D indicates that the CSTB is not fully elastic at this stage.

3. Accelerated damage stage (BC), yield failure stage of stress-strain curve. At this stage, the strain softening mechanism of CSTB begins to strengthen, and the damage value D begins to accelerate with the increase of strain.

4. Damage failure stage (DE), post-peak failure stage of stress-strain curve. At this stage, the backfill will no longer absorb energy, and the rapid increase of dissipative energy will aggravate the damage of CSTB and thus lose strength. When the energy consumption value reaches the extreme value, the damage value also reaches the maximum value, and the CSTB forms an overall failure.

Combined with the analysis results of the damage evolution process of CSTB, it can be seen that the damage evolution curve of CSTB basically shows the changing characteristics of "gentle—slow growth—rapid growth—linear growth". Moreover, at 0.2%, 0.4%, 0.6% and 0.8% fiber content, the damage extreme value of the backfill is 0.88, 0.90, 0.84 and 0.70, respectively. The damage extreme value is not 1, indicating that the CSTB still has a certain residual bearing capacity after failure.

## Effect of polypropylene fiber on failure mode of CSTB

The failure model of CSTB is shown in Fig 14. Fig 15 shows the failure characteristics of the CSTB with 3mm polypropylene fiber. The failure characteristics of the CSTB with 12mm fiber is shown in Fig 16. Fig 14 shows that the failure form of CSTB is affected by sulfur content. The CSTB shows obvious shear failure characteristics when the sulfur content is 6%, 12%, 18% and 25%, but the difference of sulfur content leads to significant differences in shear cracks on

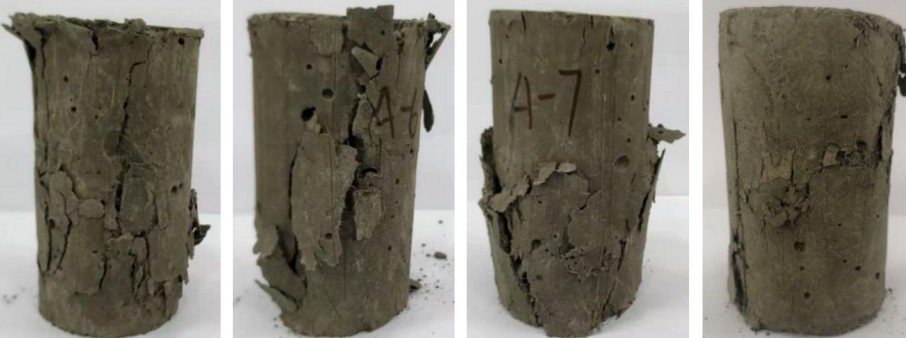

(a) 0.2% fiber content    (b) 0.4% fiber content    (c) 0.6% fiber content    (d) 0.8% fiber content

**Fig 15. Failure characteristics of the CSTB with 3mm fiber.** (a) 0.2% fiber content, (b) 0.4% fiber content, (c) 0.6% fiber content, and (d) 0.8% fiber content.

the surface of the CSTB. The surface of the CSTB showed obvious "Y-type" shear cracks when the sulfur content was 6%, 12% and 18%. For 12% and 18% sulfur content, the sample are penetrated by surface shear cracks. Once sulfur content reach 25%, there will be two "linear-type " shear cracks, and the shear cracks do not penetrate the CSTB. When the fiber content is 0.2%, there is no penetrating main crack in the CSTB sample when it is destroyed, only several cracks parallel to the axis appear on the surface of the sample and the CSTB maintains a high integrity when it is destroyed. When the fiber content is 0.4% and 0.6%, there is no penetrating main crack in the CSTB sample when it is destroyed. Despite the crack phenomenon on the surface of the CSTB, but the CSTB still maintains good integrity when it is destroyed. At 0.8% fiber content, only a few short and fine microcracks appear on the surface of the CSTB and the integrity of the CSTB is still maintained when the CSTB is destroyed. As can be seen from Fig 16, when the fiber content is 0.2%, only some small cracks appear on the surface of the CSTB sample and the CSTB maintains a good integrity when it is destroyed. When the fiber content is 0.4%, no obvious cracks exist. Although there are cracks on both sides of the CSTB sample, but the CSTB sample still maintains a good integrity after destruction. When the fiber content is 0.6% and 0.8%, there is no penetrating main crack in the CSTB sample when it is destroyed

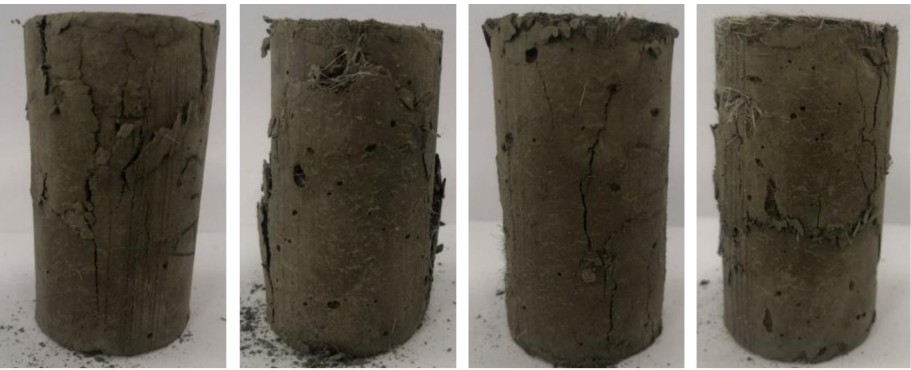

(a) 0.2% fiber content    (b) 0.4% fiber content    (c) 0.6% fiber content    (d) 0.8% fiber content

**Fig 16. Failure characteristics of the CSTB with 12mm fiber.** (a) 0.2% fiber content, (b) 0.4% fiber content, (c) 0.6% fiber content, and (d) 0.8% fiber content.

and only some small cracks appear on the CSTB surface, indicating that the CSTB also maintains a high integrity when it is destroyed. At the same time, From Figs 15 and 16, it can be seen that the integrity of the CSTB sample with a length of 12mm fiber is better than that with a length of 3mm at 0.2%, 0.4% and 0.6% fiber content, indicating that the increase in length is beneficial to the strength. Moreover, it can be seen that the fiber addition can restrain the crack growth of the CSTB sample and is beneficial for crack resistance of the CSTB.

### Effect mechanism of fiber on mechanical properties of CSTB

Fig 17 shows the microstructure characteristics of the FRCSTB. As can be seen from Fig 17, at 0.2% fiber content, obvious fiber traces appear in the cement-tailings matrix of fiber-reinforced CSTB, which illustrate that partial load can be beared by the fibers. When the polypropylene fiber is 0.4%, the fiber distribution is relatively uniform and can bear part of the load well, which is conducive to improving the strength of the CSTB [42]. Therefore, the uniform distribution of fibers should be ensured when adding fibers. When the fiber content is 0.6% and 0.8%, hydration products can be seen on the fiber surface, which illustrate that the fiber is uniform distribution state and form a bonding force with the mortar, which is beneficial for

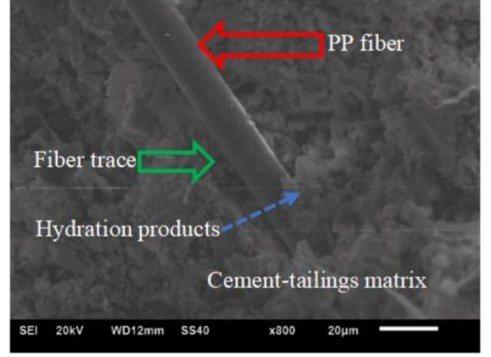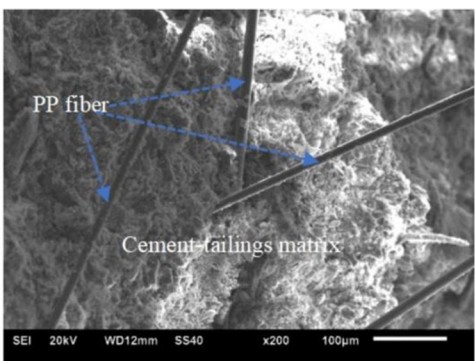

(a) 0.2% fiber content        (b) 0.4% fiber content

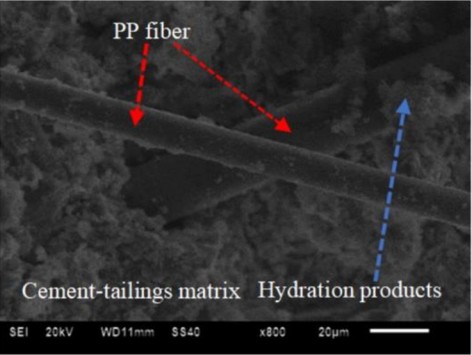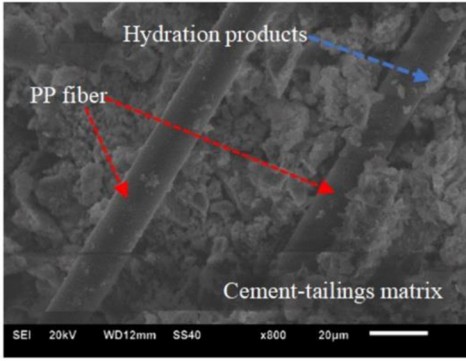

(c) 0.6% fiber content        (d) 0.8% fiber content

**Fig 17. Microstructure image of fiber reinforced CSTB.** (a) 0.2% fiber content, (b) 0.4% fiber content, (c) 0.6% fiber content, and (d) 0.8% fiber content.

the strength of the CSTB. In addition, hydration products can effectively fill the pores between fiber and cement-tailings matrix, thus further improving the bonding force between fiber and cement-tailings matrix.

## Conclusion

In this study, Mechanical properties and energy damage evolution mechanism of CSTB were systematically studied. Through the above experimental research, some research conclusions are as follows:

1. The change of sulfur content and fiber content has no effect on the form of stress-strain curve of CSTB, indicating that the CSTB with different sulfur content and fiber content are all the same kind of damage process.

2. The peak strain factor K was used to define toughness evaluation parameter. Adding appropriate amount of fiber can improve the toughness of CSTB, and the influence degree of fiber length on the toughness index of CSTB is 6mm>12mm>3mm.

3. When the fiber content is 0%~0.6%, the total strain energy of CSTB basically follows a linear growth law with the increase of fiber content, while the elastic strain energy and dissipated energy both show an exponential increase law. In the whole process of CSTB deformation and failure, the total strain energy shows the changing characteristics of increasing, the elastic strain energy shows the changing law of "gentle—linear growth—slow growth—fast decline", while the dissipated energy shows the changing law of "gentle—slow growth—fast growth—linear growth".

4. Under uniaxial loading, the damage and failure of CSTB mainly experienced four stages: initial damage, slow growth of damage, accelerated damage and damage failure. In addition, the damage evolution curve of CSTB also shows the changing characteristics of "gentle—slow growth—rapid growth—linear growth".

5. The CSTB shows obvious "Y-type" and "linear-type " shear failure characteristics during failure and the phenomenon of shear cracks penetrating the CSTB appears. No large-scale shear cracks appear in the FRCSTB. In addition, the fiber is uniform distribution state and form a bonding force with the mortar, which is beneficial for the strength of the CSTB.

## Supporting information

**S1 Data. Contains supporting datasets.**
(ZIP)

## Author Contributions

**Conceptualization:** Shenghua Yin.

**Data curation:** Wei Liu.

**Formal analysis:** Wei Liu.

**Funding acquisition:** Shenghua Yin.

**Investigation:** Wei Liu.

**Methodology:** Wei Liu.

**Project administration:** Yongqiang Hou.

**Resources:** Yongqiang Hou.

**Software:** Yongqiang Hou, Yanli Wang.

**Supervision:** Minzhe Zhang.

**Validation:** Huihui Du.

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
