## [Decision Letter · Decision Letter 0]

26 Jun 2023

PONE-D-23-17514Mechanical properties and energy damage evolution mechanism of fiber-reinforced cemented sulfur tailings backfill under uniaxial compressionPLOS ONE

Dear Dr. Yin,

Thank you for submitting your manuscript to PLOS ONE. After careful consideration, we feel that it has merit but does not fully meet PLOS ONE’s publication criteria as it currently stands. Therefore, we invite you to submit a revised version of the manuscript that addresses the points raised during the review process.

We look forward to receiving your revised manuscript.

Kind regards,

Khalil Abdelrazek Khalil, Ph.D.

Academic Editor

PLOS ONE

Journal Requirements:

   "This work was supported the Shandong Provincial Major Science and Technology Innovation Project, China (No. 2019SDZY05), the Key Program of National Natural Science Foundation of China (Grant No. 51734001), the Key Program of National Natural Science Foundation of China（52034001）."

   "This work was supported the Shandong Provincial Major Science and Technology Innovation Project, China (No. 2019SDZY05), the Key Program of National Natural Science Foundation of China (Grant No. 51734001), the Key Program of National Natural Science Foundation of China（52034001）."

   "This work was supported the Shandong Provincial Major Science and Technology Innovation Project, China (No. 2019SDZY05), the Key Program of National Natural Science Foundation of China (Grant No. 51734001), the Key Program of National Natural Science Foundation of China（52034001）."

6. Please ensure that you refer to Figure 12 in your text as, if accepted, production will need this reference to link the reader to the figure.

Reviewers' comments:

Reviewer's Responses to Questions

**Comments to the Author**

1. Is the manuscript technically sound, and do the data support the conclusions?

Reviewer #1: Yes

Reviewer #2: Yes

2. Has the statistical analysis been performed appropriately and rigorously? 

Reviewer #1: Yes

Reviewer #2: Yes

3. Have the authors made all data underlying the findings in their manuscript fully available?

Reviewer #1: Yes

Reviewer #2: Yes

4. Is the manuscript presented in an intelligible fashion and written in standard English?

Reviewer #1: Yes

Reviewer #2: Yes

5. Review Comments to the Author

Reviewer #1: This paper is of value in some sense, and it can be accepted by this journal if the following aspects is considered carefully

1. The review of the background should be polished, and some latest relative references should be cited

2. Some comaprisons with the existing results should be added.

Reviewer #2: This paper aims to investigate the mechanical properties and energy damage evolution mechanism of fiber-reinforced cemented sulfur tailings backfill. The research content of this paper is novel, and the research results will provide great help for the development of tailings backfill material. However, there are still some problems in the manuscript that require the authors to give a reasonable explanation, as follows:

1. Generally, the introduction does not specifically highlight the research necessity of energy evolution characteristic

of the tailings backfill material.

2. Section 2.2, please explain the reason for choosing the cement-to-tailings (c/t) ratio (i.e., 1:8) and the fiber length (3mm, 6mm and 12mm).

3. The curing time of concrete is 28d, is this curing time suitable for the fiber-reinforced cemented sulfur tailings backfill in the paper?

4. In Fig. 3, the maximum fiber length is 12mm, which is larger than 1/10 of the diameter of the sample, how to ensure the uniformity of these mixed materials?

5. In Fig.9, the tension and compression ratio of CSTB does not monotonically change with the increase of the fiber length, necessary explanations should be added.

6. In Fig.16, it is difficult to find the difference in fiber content of the materials shown in the figure.

7. In Section 1, the seepage property is also very important to the utilization of CSTB, to support the authors' agreement, it is suggested to add more references, such as: (1) Doi: 10.1016/j.tust.2022.104525 (2) Doi: 10.1007/s00603-022-02878-9 (3) Doi: 10.1007/s40789-022-00525-w

6. PLOS authors have the option to publish the peer review history of their article (what does this mean?). If published, this will include your full peer review and any attached files.

Reviewer #1: No

Reviewer #2: No

---

## [Author Response · Author response to Decision Letter 0]

29 Jul 2023

To editor,

1. The manuscript has been formatted according to the template provided by the journal.

2. The funding-related text has been deleted. And No update is required for the funding information.

4. The data that support the findings of this study are available from the corresponding author, upon reasonable request.

5. The ORCID iD has added for the corresponding author.

6. Due to an oversight, Figure 12 was not mentioned in the main text of the original manuscript. In the revised version, all figures are appropriately referenced in the main text.

To the Reviewer 1:

Reviewer 1: 

This paper is of value in some sense, and it can be accepted by this journal if the following aspects is considered carefully

1. The review of the background should be polished, and some latest relative references should be cited

Answer: The introduction section has been polished, and some recent literature has been added as well. See line 47-56. The specisic references are as follows:

18. Wang YY, Yu ZQ, Wang HW. Experimental investigation on some performance of rubber ﬁber modiﬁed cemented paste backﬁll. Constr Build Mater. 2021; 271: 121586.

19. Xue GL, Yilma E. Strength, acoustic, and fractal behavior of ffber reinforced cemented tailings backffll subjected to triaxial compression loads. Constr Build Mater. 2022; 338: 127667.

20. Xue GL, Gu YT, Zhang X, Wu SJ, Sun PC, Cui JQ, et al. Mechanical behavior and microscopic mechanism of fiber reinforced coarse aggregate cemented backfill. Constr Build Mater. 2023; 366: 130093.

21. He W, Liu L, Fang ZY, Gao YH, Sun WJ. Effect of polypropylene ffber on properties of modiffed magnesium-coal-based solid waste backffll materials. Constr Build Mater. 2023; 362: 129695.

2. Some comaprisons with the existing results should be added.

Answer: The comparative analysis was conducted between the research findings of this study and the existing literature. See line 233-241 and Table 5.

To the Reviewer 2:

Reviewer 2: This paper aims to investigate the mechanical properties and energy damage evolution mechanism of fiber-reinforced cemented sulfur tailings backfill. The research content of this paper is novel, and the research results will provide great help for the development of tailings backfill material. However, there are still some problems in the manuscript that require the authors to give a reasonable explanation, as follows:

1. Generally, the introduction does not specifically highlight the research necessity of energy evolution characteristic of the tailings backfill material.

Answer: The necessity of studying the energy evolution in tailings backfill is further elucidated in the introduction. See line 28-30.

2. Section 2.2, please explain the reason for choosing the cement-to-tailings (c/t) ratio (i.e., 1:8) and the fiber length (3mm, 6mm and 12mm).

Answer: In accordance with the requirements, the reasons for selecting tailings mix proportion parameters and fiber length are supplemented in original Section 2.2. See line 117-118.

3. The curing time of concrete is 28d, is this curing time suitable for the fiber-reinforced cemented sulfur tailings backfill in the paper?

Answer: During the mining process in a mine, the backfill material in the stope can generally be excavated after a curing period of 28 days. Therefore, a 28-day curing time is considered reasonable and reliable.

4. In Fig. 3, the maximum fiber length is 12mm, which is larger than 1/10 of the diameter of the sample, how to ensure the uniformity of these mixed materials?

Answer: During the mixing process, the fibers were added to the dry materials using a "dry mixing method". This involved adding the fibers in small quantities multiple times during the mixing of the dry materials to ensure their thorough dispersion. After achieving proper fiber dispersion, water was added and the mixture was further stirred for 3 to 5 minutes to prepare a uniformly mixed backfill slurry.

5. In Fig.9, the tension and compression ratio of CSTB does not monotonically change with the increase of the fiber length, necessary explanations should be added.

Answer: As requested, the explanation has been provided in the manuscript as required. See line 291-294.

6. In Fig.16, it is difficult to find the difference in fiber content of the materials shown in the figure.

Answer: The original manuscript section 3.6 primarily investigates the influence mechanism of fibers on the mechanical properties of the backfill material. Therefore, special emphasis is placed on analyzing the bond between fibers and the mortar matrix as well as the hydration products. The relationship between the distribution state of fibers within the specimens and the fiber content will be addressed in future research as a next step.

7. In Section 1, the seepage property is also very important to the utilization of CSTB, to support the authors' agreement, it is suggested to add more references, such as: (1) Doi: 10.1016/j.tust.2022.104525 (2) Doi: 10.1007/s00603-022-02878-9 (3) Doi: 10.1007/s40789-022-00525-w

Answer: As requested, the importance of permeability as a significant characteristic of the backfill material has been addressed in the introduction, and relevant literature suggested by experts has been cited. See line 28-30 and 521-527.

---

## [Decision Letter · Decision Letter 1]

15 Aug 2023

Mechanical properties and energy damage evolution mechanism of fiber-reinforced cemented sulfur tailings backfill under uniaxial compression

PONE-D-23-17514R1

Dear Dr. Yin,

We’re pleased to inform you that your manuscript has been judged scientifically suitable for publication and will be formally accepted for publication once it meets all outstanding technical requirements.

Kind regards,

Khalil Abdelrazek Khalil, Ph.D.

Academic Editor

PLOS ONE

Additional Editor Comments (optional):

Reviewers' comments:

Reviewer's Responses to Questions

**Comments to the Author**

1. If the authors have adequately addressed your comments raised in a previous round of review and you feel that this manuscript is now acceptable for publication, you may indicate that here to bypass the “Comments to the Author” section, enter your conflict of interest statement in the “Confidential to Editor” section, and submit your "Accept" recommendation.

Reviewer #1: All comments have been addressed

Reviewer #2: All comments have been addressed

2. Is the manuscript technically sound, and do the data support the conclusions?

Reviewer #1: Yes

Reviewer #2: Yes

3. Has the statistical analysis been performed appropriately and rigorously? 

Reviewer #1: Yes

Reviewer #2: Yes

4. Have the authors made all data underlying the findings in their manuscript fully available?

Reviewer #1: Yes

Reviewer #2: Yes

5. Is the manuscript presented in an intelligible fashion and written in standard English?

Reviewer #1: Yes

Reviewer #2: Yes

6. Review Comments to the Author

Reviewer #1: his paper studies mechanical properties and energy damage evolution of fiber-

reinforced cemented sulfur tailings (CSTB) backfill. The effects of fiber length and fiber

content on the stress, toughness and failure properties of the CSTB were

systematically revealed. In addition, the energy index evolution law was studied, and

the energy damage evolution mechanism of CSTB was revealed. The results show that

the deformation failure of fiber-reinforced CSTB mainly goes through four stages: initial

crack compaction, linear elastic deformation, yield failure and post-peak failure.

The revised version is good and suitable for publication

Reviewer #2: The authors have addressed all my comments very well. The manuscript can be accepted in its present version.

7. PLOS authors have the option to publish the peer review history of their article (what does this mean?). If published, this will include your full peer review and any attached files.

Reviewer #1: No

Reviewer #2: No

---

## [Editor Report · Acceptance letter]

22 Aug 2023

PONE-D-23-17514R1 

Mechanical properties and energy damage evolution mechanism of fiber-reinforced cemented sulfur tailings backfill under uniaxial compression 

Dear Dr. Yin:

I'm pleased to inform you that your manuscript has been deemed suitable for publication in PLOS ONE. Congratulations! Your manuscript is now with our production department. 

Kind regards, 

on behalf of

Dr. Khalil Abdelrazek Khalil 

Academic Editor

PLOS ONE